# The Yin and Yang of Heartbeats: Magnesium–Calcium Antagonism Is Essential for Cardiac Excitation–Contraction Coupling

**DOI:** 10.3390/cells14161280

**Published:** 2025-08-18

**Authors:** Chiara Marabelli, Demetrio J. Santiago, Silvia G. Priori

**Affiliations:** 1Department of Molecular Medicine, University of Pavia, 27100 Pavia, Italy; 2Laboratory of Molecular Cardiology, IRCCS ICS Maugeri, 27100 Pavia, Italy; 3Centro Nacional de Investigaciones Cardiovasculares Carlos III (CNIC), 28029 Madrid, Spain; demetriojulian.santiago@cnic.es

**Keywords:** magnesium, calcium signaling, cardiac electrophysiology, excitation–contraction coupling, calcium-induced calcium release, ion channels

## Abstract

While calcium (Ca^2+^) is a universal cellular messenger, the ionic properties of magnesium (Mg^2+^) make it less suited for rapid signaling and more for structural integrity. Still, besides being a passive player, Mg^2+^ is the only active Ca^2+^ antagonist, essential for tuning the efficacy of Ca^2+^-dependent cardiac excitation–contraction coupling (ECC) and for ensuring cardiac function robustness and stability. This review aims to provide a comprehensive framework to link the structural and molecular mechanisms of Mg^2+^/Ca^2+^ antagonistic binding across key proteins of the cardiac ECC machinery to their physiopathological relevance. The pervasive “dampening” effect of Mg^2+^ on ECC activity is exerted across various players and mechanisms, and lies in the ions’ physiological competition for multiple, flexible binding protein motifs across multiple compartments. Mg^2+^ profoundly modulates the cardiac action potential waveform by inhibiting the L-type Ca^2+^ channel Cav1.2, i.e., the key trigger of cardiac ryanodine receptor (RyR2) opening. Cytosolic Mg^2+^ favors RyR2 closed or inactive conformations not only through physical binding at specific sites, but also indirectly through modulation of RyR2 phosphorylation by Camk2d and PKA. RyR2 is also potently inhibited by luminal Mg^2+^, a vital mechanism in the cardiac setting for preventing excessive Ca^2+^ release during diastole. This mechanism, able to distinguish between Ca^2+^ and Mg^2+^, is mediated by luminal partners Calsequestrin 2 (CASQ2) and Triadin (TRDN). In addition, Mg^2+^ favors a rearrangement of the RyR2 cluster configuration that is associated with lower Ca^2+^ spark frequencies.

## 1. Introduction

The role of calcium (Ca^2+^) as a universal second messenger cannot be overstated. Ca^2+^ is omnipresent, involved in processes spanning from the shortest time scales (e.g., vesicular fusion in the µs range) to the years-long occurrence of diseases (e.g., heart failure) [1,2]. Cells go to great lengths in order to contain, store, and profit from the signaling actions of Ca^2+^: Ca^2+^ buffers serve as contention means, organelles such as the sarco-endoplasmic reticulum and mitochondria serve as Ca^2+^ storage areas, and Ca^2+^-sensing proteins are used for local and global signaling [1,3,4]. Hormones carefully regulate extracellular Ca^2+^ levels in the larger organismal context.

The above mechanisms, however, only contextualize and address the signaling actions of Ca^2+^, without truly tuning it. It is magnesium (Mg^2+^), a relatively similar cation in appearance, that allows modulation of the amplitude of Ca^2+^-dependent processes by antagonizing them. By opposing and competing with each other, Ca^2+^ and Mg^2+^ successfully defeat dangerous increases in cellular entropy [4,5,6,7,8].

This review focuses on the roles played by Mg^2+^ in cardiac excitation–contraction coupling (ECC), a Ca^2+^-controlled process by which the heartbeat occurs in the millisecond time scale. We will start by discussing the ionic nature of Mg^2+^ as compared to Ca^2+^, and how their differences are employed by biological systems to elicit distinct and antagonistic mechanisms. We will further explore the structural characteristics of Ca^2+^- and Mg^2+^-binding sites in different cardiac proteins, and dissect how the most delicate Ca^2+^-dependent ECC mechanisms are also sensitive to Mg^2+^. In between, we will briefly touch upon those Mg^2+^-related aspects that concern heart pathophysiology. The organismal and cellular regulations of Mg^2+^ levels are outside the scope of this review: the reader is referred to [9,10] for excellent discussions. By examining the intricate interplay between Mg^2+^ and Ca^2+^ in cardiac EC coupling, this review seeks to provide a comprehensive understanding of Mg^2+^’s crucial modulatory role, from its molecular binding sites and their selectivity to its profound clinical implications.

## 2. The Basis of the Antagonistic Actions by Ca^2+^ and Mg^2+^

Mg^2+^ is the second most abundant intracellular cation in cardiomyocytes, with a total concentration about 10 mM [11], of which the free Mg^2+^ concentration is tightly maintained in the 0.5 to 1 mM range [1,5,12,13,14]. Exactly because of the very small fluctuations in free Mg^2+^ abundance, studies on cellular dynamic processes have mainly focused on the orders of magnitude larger variation in Ca^2+^ concentrations, largely obscuring the Mg^2+^-dependent fine adjustments of these same processes, including cardiac homeostasis [10,15]. Despite both being divalent cations, fundamental differences in their ionic properties shape their distinct, often competitive, interactions with various proteins. The Ca^2+^ ion features a larger ionic radius (1.05 Å) than that of Mg^2+^ (0.78 Å). This implies a lower charge density and consequently a lower desolvation energy. Together with the easier exchange of ligand partners, the variable coordination number of Ca^2+^ (between 6 and 9) allows it to be easily grabbed by solvent-exposed protein motifs within geometrically flexible coordination complexes [16]. In contrast, free Mg^2+^ is usually present within a stable hydration shell, shaped by the rigid octahedral disposition of six water molecules, the removal of which is energetically very demanding [16,17]. In addition, it is not unusual for this ion to have additional water layers around it, so that changes of up to 400 times in volume have been reported between hydrated and non-hydrated Mg^2+^, compared to a ~25-fold difference for Ca^2+^ [18,19]. This increase in radius, unlike calcium, prevents magnesium from passing through narrow ion channels. Indeed, few Mg^2+^ transporters are known.

The above differences between Ca^2+^ and Mg^2+^ dictate the structure of their protein binding sites and their functional effects in biological systems. The flexibility of Ca^2+^ coordination within proteins acquires a functional role in processes such as sarcomere contraction, opening of transmembrane channels, and activation of signaling enzymes. Ca^2+^ binding occurs at flexible and adaptable protein motifs, such as the EF-hand motif (found in cardiac troponin C (cTnC) and calmodulin (CaM)), and is coordinated by mixed ligand types (e.g., aspartate, glutamate, asparagine, and carbonyl oxygens) [16,20,21].

On the other hand, the tight coordination of Mg^2+^ atoms is usually shaped by the octahedral disposition of six hard oxygens from carboxyl groups (Asp, Glu) and/or phosphates (ATP, GTP, etc.), with a lower variability in the distances between cation and oxygen atom compared to Ca^2+^ (2.05 to 2.25 Å for Mg^2+^ vs. 2.2 to 2.7 Å for Ca^2+^) [19]. The energetically expensive and structurally rigid binding of Mg^2+^ is better suited for the structural integrity of proteins (this implication is supported by 40% more proteins containing Mg^2+^ vs. Ca^2+^ atoms, as published in the RCSB Protein Data Bank [22]). Mg^2+^ is also a structural cofactor for ATP-dependent enzymes like kinases and ATPases, whereby it facilitates catalysis by stabilizing transition states or providing a binding site for substrates.

However, the distinction between the architectures of protein Ca^2+^- and/or Mg^2+^- binding sites is not strictly binary. Many protein sites bind both cations, sometimes even in the same buffer condition [21,23]. The ground of the competition between Mg^2+^ and Ca^2+^ ions occurs exactly within this spectrum of protein divalent-binding sites, mainly at the larger and more flexible motifs rather than at the rigid and smaller ones evolutionarily designed for Mg^2+^ binding. In addition to the binding site design, the protein dynamics and the ionic environment also shape the relative binding equilibria of Mg^2+^ and Ca^2+^. The functional implication is that Mg^2+^, whose intracellular concentration oscillates far less than that of Ca^2+^, acts as a natural competitive “damping agent” of Ca^2+^-dependent events. When transient spikes of Ca^2+^ occur, Mg^2+^ is displaced and Ca^2+^-dependent signaling mechanisms are unlocked.

## 3. Ca^2+^-Driven ECC Runs Across Multiple Cellular Landscapes

Cardiac ECC is the fundamental Ca^2+^-dependent process by which an electrical stimulus, the action potential (AP), is translated into coordinated contraction of the myocardial cells. As finely tuned, Ca^2+^-dependent interconnections between sequential stages occurs in distinct cellular compartments, not only the dynamics of Ca^2+^-handling proteins, but also their precise spatial arrangement, are extremely critical to ECC [1,24].

### 3.1. Structural Aspects of Cardiac ECC

ECC occurs in specialized structures within myocytes known as “dyads”, formed by periodically or near-periodically juxtaposed areas of the sarcolemma or its invaginations, the transverse (T-) tubules, and highly specialized domains of the sarcoplasmic reticulum known as the “junctional” sarcoplasmic reticulum (jSR). The juxtaposition of both organelles forms a narrow (≤20 nm) [25], protein-rich, and diffusionally constrained environment known as the “dyadic space”. Certain proteins, such as Junctophilin 2 (JPH2), are capable of structurally linking the two organelles by dwelling within both membranes simultaneously.

### 3.2. Functional Aspects of Cardiac ECC

ECC is initiated by the arrival of an AP at the dyadic sarcolemma (i.e., by a rapid, transient change in the voltage across the membrane). This electrical signal triggers the opening of voltage-sensitive, transmembrane L-type Ca^2+^ channels (Cav1.2) [24,26], causing a minor influx of extracellular Ca^2+^ into the narrow dyadic space. Here, due to diffusional constriction [25], a few Ca^2+^ ions can translate into relatively high concentrations and rapidly activate clusters of jSR-embedded, Ca^2+^-sensitive Ca^2+^ release channels (known as ryanodine receptors type 2; RyR2s) [13,27,28]. This cytosolic Ca^2+^-dependent RyR2 activation, known as cell-wide Ca^2+^-induced Ca^2+^ release (CICR), leads to a substantial outflow of Ca^2+^ through the RyR2 channels from the jSR lumen into the cytosol [24,27,29], which is in turn used for cellular contraction plus other regulatory pathways (depending on heart rate, metabolic demands, pathophysiological status, etc.). The cytosolic Ca^2+^ concentration rises from a diastolic (resting if cell not stimulated) 100 nM value to peaks of 1 μM during systole [1].

CICR does occur stochastically and locally at low frequency in cells (the so-called “Ca^2+^ sparks”), which occurs when the statistically improbable openings of nearby Cav1.2s and/or RyR2s synchronize and the influx of Ca^2+^ towards the dyadic space triggers the opening of neighboring RyR2s.

Mg^2+^ acts as an inhibitor of this CICR process, both by competitive inhibition of Ca^2+^-dependent activation sites and by inhibiting RyR2 by itself at Ca^2+^/Mg^2+^-dependent inactivation sites [30] (see below sections). To understand the relevance of Mg^2+^ inhibition to CICR in physiological conditions, it must be pointed out that, out of the (up to) 100 RyR2s present in a murine RyR2 cluster, on average only 9-11 RyR2s activate at the peak of a murine Ca^2+^ spark [31]. Besides the effects of Mg^2+^, additional mechanisms further regulate the likeliness of CICR in major ways, such as RyR2 fragmentation into subclusters [32], the presence/absence/mutation of Junctophilin 2 [33], dyadic buffering [34], and free levels of jSR Ca^2+^ [35].

In the presence of a cardiac AP, depolarization opens many Cav1.2s, the Ca^2+^ stimulus becomes persistent, Mg^2+^ is displaced by Ca^2+^ at the RyR2 Ca^2+^-dependent activation site, and Ca^2+^ sparks synchronize to produce the whole-cell Ca^2+^ transient. In certain pathophysiological states (e.g., hypomagnesemia, arrhythmias, heart failure, SR Ca^2+^ overload), or when the sarcolemma is damaged, untimely Ca^2+^ release may occur in the absence of an AP. Such CICR is sequential and fueled by a “fire–diffuse–fire” mechanism: Ca^2+^ is spontaneously released from existing dyads, and it diffuses and triggers CICR at nearby dyads without Cav1.2 involvement. This “fire–diffuse–fire” also occurs physiologically during systole in instances where the association between T-tubules and jSR is not present (e.g., at the center of atrial cells and iPSC-derived cardiomyocytes) or has been lost (failing ventricular myocytes), the latter at the cost of cardiac output [36].

For completeness, we must add that ECC comprises two signaling pathways: anterograde Cav1.2-to-RyR2 signaling, which leads to CICR (discussed above); and retrograde RyR2-to-Cav1.2 signaling (via Ca^2+^ release), which leads to Cav1.2 Ca^2+^-dependent inactivation. This retrograde pathway contributes to AP repolarization and prevents cells from overloading with Ca^2+^.

Furthermore, the link between CICR, Cav1.2 inactivation, SERCA-mediated uptake, and NCX-mediated extrusion of cytosolic Ca^2+^ constitutes the basis of so-called “Ca^2+^ autoregulation”, which develops in time frameworks of several beats and homeostatically maintains cardiac output. Autoregulation also explains why Ca^2+^-handling-dependent arrhythmias do not occur at baseline situations but instead occur during beta-adrenergic stimulation [37]. While the Mg^2+^-dependent effects on the aforementioned pathways will be discussed below, we know of no studies specifically focused on the overall Mg^2+^ regulation of Ca^2+^ autoregulation, and therefore we will no longer focus on this aspect of ECC.

### 3.3. Mg^2+^ Shapes the Kinetics of Contraction and Relaxation

During individual Ca^2+^ sparks, the amount of non-diffusible Ca^2+^-binding partners between dyads and myofilaments is so large that it limits the increase in free Ca^2+^ to sub-micron distances from the release sites. Ca^2+^ diffusion is supported by ATP, a highly mobile and abundant (up to 5 mM in cardiac cells) Ca^2+^ buffer, with very fast on–off kinetics for both Ca^2+^ and Mg^2+^. ATP avidly binds some of the released Ca^2+^ (therefore releasing Mg^2+^ bound during diastole) and facilitates the trip of complexed Ca^2+^ towards the contractile filaments [38]. Additionally, spatiotemporal summation of sparks during whole-cell ECC allows more Ca^2+^ to reach the myofilaments.

Once at the contractile elements and when in free form, Ca^2+^ binds to cardiac troponin C (cTnC). As cytosolic Ca^2+^ displaces Mg^2+^ in binding to cTnC, it induces a conformational change in the troponin–tropomyosin complex, which in turn exposes the actin–myosin binding sites and allows sarcomere contraction [21,39], which itself depends on Mg-ATP.

For relaxation to occur, Ca^2+^ is removed from the cytosol primarily by the Sarco/Endoplasmic Reticulum Ca^2+^-ATPase (SERCA), which populates at high density the membrane of the non-junctional sarcoplasmic reticulum [24]. This Ca^2+^ removal mechanism, once again, depends on Mg-ATP expenditure. Secondly, the sodium–calcium exchanger (NCX), located in the sarcolemma (yet excluded from the immediate dyadic junctions), provides bulk Ca^2+^ extrusion from the cardiomyocyte into the extracellular space, in a Mg^2+^-dependent manner (see below).

Lastly, as mentioned above, part of the released Ca^2+^ is used for non-ECC purposes. The mitochondrial calcium uniporter (MCU), generally lying in the outer mitochondrial membrane in very close proximity to the jSR, contributes to Ca^2+^ homeostasis while triggering intra-mitochondrial Ca^2+^-signaling cascades for the stimulation of oxidative metabolism [1,15,40,41,42]. This jSR–mitochondrial coupling is important mechanistically, as it allows the heart to adjust to different heart rates. Notice that at increased heart rates, Ca^2+^ release occurs more often, so Ca^2+^ enters mitochondria more frequency and more ATP can be produced to match the increases in Mg-ATP expenditure for both the contraction and recycling of ions. That said, Ca^2+^ is only able to increase respiration by a factor of ~2, where changes in left ventricular filling can increase respiration more than 10-fold, so this ECC-related mechanism is still a small contributor to myocardial energetics [43].

## 4. Mg^2+^/Ca^2+^ Crosstalk Across ECC Key Cytosolic Compartments

The vital competition between Mg^2+^ and Ca^2+^ during ECC occurs at distinct timings and spaces: the cytosol, the narrow dyadic space, and the jSR lumen. The concentration distribution of each ion across distinct cellular compartments, and the ion-binding properties and dynamics of the residing proteins, dictate the functional outcome of their crosstalk [1,24]. The overall ECC process thus relies on very subtle differences in Ca^2+^ and Mg^2+^ handling and signaling across compartments, up to the point that dysregulation of a single component in a single cellular location can affect the entire process.

In the cytosol, both during diastole/at rest and during contraction, the concentration of free Mg^2+^ (in the 0.8–1 mM range) is orders of magnitude higher than that of cytoplasmic Ca^2+^, fluctuating from 100 nM to approximately 1 µM during systole [1,10,24]. This vast numerical advantage endows Mg^2+^ with a large advantage in competing with Ca^2+^ at shared binding sites on various proteins [4,10,15]. More specifically, Mg^2+^ functionally antagonizes Ca^2+^ at sites critical for ionic channels [12,26,27]. In addition, a large portion (about 5 mM) of intracellular Mg^2+^ is retained through the binding of ATP [26,44,45]. The homeostasis of Mg^2+^ is indeed intimately linked to ATP levels: depletion of ATP, exacerbated by conditions such as alcohol exposure, can lead to impaired Mg^2+^ retention within mitochondria and the cytoplasm [46]. Conversely, this binding also ensures the availability of Mg-ATP, the physiologically active form of ATP [1,17,47].

### 4.1. Effects on Cellular Excitability

The sarcolemma is equipped with a battery of ion channels and transporters that regulate the various ionic currents of the cardiac AP (Figure 1A). Despite being often neglected, Mg^2+^ profoundly modulates the AP waveform through its intricate interactions with sarcolemmal ion channels (Figure 1B), often through direct and specific mechanisms [15,48] but also through action on other protein interactors such as G-proteins [48]. In addition, extracellular Mg^2+^ and Ca^2+^ both screen the negative charges on the outside of the sarcolemma and channel proteins. This screening effect confers a proper voltage-dependent behavior to ion channels, affecting how they open and inactivate [49].

#### 4.1.1. Main Depolarizing (Inward) Currents

The sodium current (INa) is mediated by voltage-gated Na^+^ channels (mainly Nav1.5), and is responsible for the rapid sarcolemmal depolarization (phase 0, Figure 1)). Evidence in ventricular cells suggests Mg^2+^ primarily exerts an open-channel blocking effect on outward currents in a concentration- and voltage-dependent manner, with limited direct allosteric modulation [50]. The biological implications of such Mg^2+^-dependent regulation, if any, remain unknown, as there is no outward INa in cardiac cells in physiological conditions.

The L-type Ca^2+^ current (ICaL), mediated by Cav1.2s, is the principal maintainer of the AP plateau (phase 2, Figure 1) and the provider of the trigger Ca^2+^ influx (ICaL) that initiates cardiac ECC. Its high sensitivity to intracellular Mg^2+^ concentration has been more deeply studied than that of many other channels, revealing that Mg^2+^ exerts both direct and indirect inhibitory effects [4,7,51]. Structural and mutagenesis studies indicate that Mg^2+^ binding to an EF-hand motif in the C-terminal domain of Cav1.2 reduces peak current amplitude and enhances voltage-dependent inactivation (VDI) [13,52,53]. During myocardial ischemia, where ATP hydrolysis leads to a rise in cytosolic Mg^2+^ to 2.1–2.3 mM [53,54], Mg^2+^-dependent suppression of ICaL shortens AP duration, mitigating both Ca^2+^ overload and cytotoxicity. Conversely, in pathological conditions like heart failure, low Mg^2+^ levels contribute to an adaptive increase in ICaL, increasing the susceptibility to arrhythmias and impaired relaxation [52,54,55]. Moreover, Mg^2+^ antagonizes the β-adrenergic receptor (β-AR)–PKA pathway both by reducing the affinity of PKA for activating Ca^2+^ and by limiting the structural flexibility of the CaV1.2 cytoplasmic domain, hence the availability of exposed serines as substrates for PKA [8,13].

#### 4.1.2. Main Repolarizing (Outward) Currents

The effects of Mg^2+^ in the transient outward current (Ito), which is responsible for the AP hump (phase 1, Figure 1), have not been widely studied in humans, where the current is mainly mediated by Kv4.3 channels, with some contribution by Kv4.2. In rats, where this current is mainly mediated by Kv4.2, Mg^2+^ deficiency was shown to produce a decrease in the levels of Kv4.2 mRNA and protein, leading to corresponding decreases in Ito [56].

Delayed rectifier K^+^ currents (IKr, IKs) are also critical repolarizing currents, defining the duration of the AP plateau (phase 2) and phase 3 (Figure 1). IKr studies in nominally Mg^2+^-free intracellular solution suggest that Mg^2+^ does not block the pore of the HERG channel [57]. Additional studies suggest that intracellular Mg^2+^ above 10 μM is needed to maintain repolarization reserve [58].

The inward rectifier K^+^ current (IK1), mediated by Kir2.1 channels, is fundamental for enabling rapid membrane repolarization during phase 3 of the AP and for maintaining a stable resting membrane potential during phase 4 (Figure 1). During the plateau of the cardiac AP (phase 2), the membrane potential is more positive on the inside of the sarcolemma than on its outside, creating a local driving force for Mg^2+^ ions towards the pore of open Kir2.1 channels. This Mg^2+^ binds the pore, leading to voltage-dependent open-channel block and causing a characteristic inward rectification (i.e., block of outward K^+^ currents at depolarized potentials) [4,7,48,59]. At membrane potentials progressively closer to the resting membrane potential (i.e., phases 3 and 4 of the AP), Mg^2+^ ions progressively abandon Kir2.1 pores, allowing a normal flow of K^+^ ions, which repolarizes the membrane and maintains the resting membrane potential. Mg^2+^ block is remarkably instantaneous, and involves the Ser165 and Asp172 residues in the transmembrane domain (TM2), whereas Glu224 and Glu299 at the cytoplasmic bundle-crossing region determine the Mg^2+^ sensitivity [59]. Experimentally, it has been shown that Mg^2+^ deficiency can cause transcriptional downregulation of the amounts of Kir2.1 mRNA and protein. Conversely, Mg^2+^ overload upregulates the amounts of Kir2.1 [56]. An in silico study specifically analyzed the effects of hypomagnesemia on IK1 rectification while maintaining constant Kir2.1 levels and in the absence of hypomagnesemia-related changes to other ionic currents. The study concluded that a lack of Mg^2+^ in these simulated conditions would unexpectedly decrease outward IK1 currents, leading to prolonged AP duration (see Figure 5 therein), which in turn could result in increased risk of arrhythmias (early afterdepolarizations, heterogeneity of AP duration) [60].

The Na^+^/K^+^ ATPase current (INaK) maintains the resting membrane potential and repolarizes the AP (Figure 1B) [61]. Mg^2+^ deficiency contributes to cardiac arrhythmia by disrupting the activity of this ATPase due to lower Mg-ATP levels [62].

#### 4.1.3. Other Repolarizing Currents

The muscarinic K^+^ current (IK, ACh, Kir3.1) is predominantly found in atrial and nodal cells, controlling diastolic potential and pacemaker depolarization [48]. Similarly to IK1, the inward rectification of IKACh is fundamentally due to a voltage-dependent block by internal Mg^2+^. However, the voltage-dependent sensitivity of this block differs significantly from that of IK1, implying unique Mg^2+^-binding site properties [7].

ATP-Sensitive K^+^ (KATP, Kir6.2) currents link cellular metabolic state to excitability via a linear current–voltage dependency. As the Kd for Mg^2+^ block is voltage-dependent, ranging from approximately 10 mM at −60 mV to 0.5 mM at +40 mV, Mg^2+^-dependent inhibitory effects are also voltage-dependent. In addition, MgADP binding activates the channel by counteracting Mg-ATP inhibition [7,48].

#### 4.1.4. Na^+^/Ca^2+^ Exchange

The NCX-dependent current (INCX) can behave as either a depolarizing or a repolarizing current depending on the specific timing within the AP and ionic conditions. Early into the action potential, INCX brings a tiny amount of Ca^2+^ inside cells, whereas during most of the AP and diastole it removes Ca^2+^ from the cytosol. Despite this primary importance for cell relaxation and Ca^2+^, only a few studies have addressed the effects of intracellular Mg^2+^ on INCX. NCX activity is strongly increased by intracellular calcium, at a specific intracellular allosteric regulatory site. The competition between Mg^2+^ and Ca^2+^ for binding to this site suggests that low Mg^2+^ levels may promote arrhythmogenesis and cellular Ca^2+^ loss (e.g., heart failure) via NCX dysregulation [6,48,63].

### 4.2. Effect on Cellular Contractile Response

In addition to its structural presence at sites III and IV of cardiac troponin C (cTnC), physiological amounts of Mg^2+^ compete with Ca^2+^ for binding at the regulatory site II of cTnC [21,64]. This specific interaction impacts the Ca^2+^ sensitivity of every single myofilament and thus the contractile force generated by the overall muscle. In hypertrophic cardiomyopathy, mutations in cTnC alter the protein’s affinity for both Ca^2+^ and Mg^2+^, disrupting contractile regulation [21].

### 4.3. Effect on Relaxation

SERCA (more specifically SERCA2a in the cardiac muscle) is the workhorse of cardiac relaxation, actively pumping Ca^2+^ from the cytosol back into the SR. Mg^2+^ is essential for coordinating ATP during the phosphorylation step of the pump cycle [16,65]. Molecular dynamics simulations have shown that Mg^2+^ can compete with Ca^2+^ for binding to SERCA’s canonical Ca^2+^-binding sites (sites I and II) [65]. Other effects of Mg^2+^ on SERCA are expectedly mediated by the various Mg-ATP-dependent kinases, acetylases, and SUMOylases that control its activity and that of its main inhibitory partner, phospholamban (PLB) [66,67].

**Figure 1 cells-14-01280-f001:**
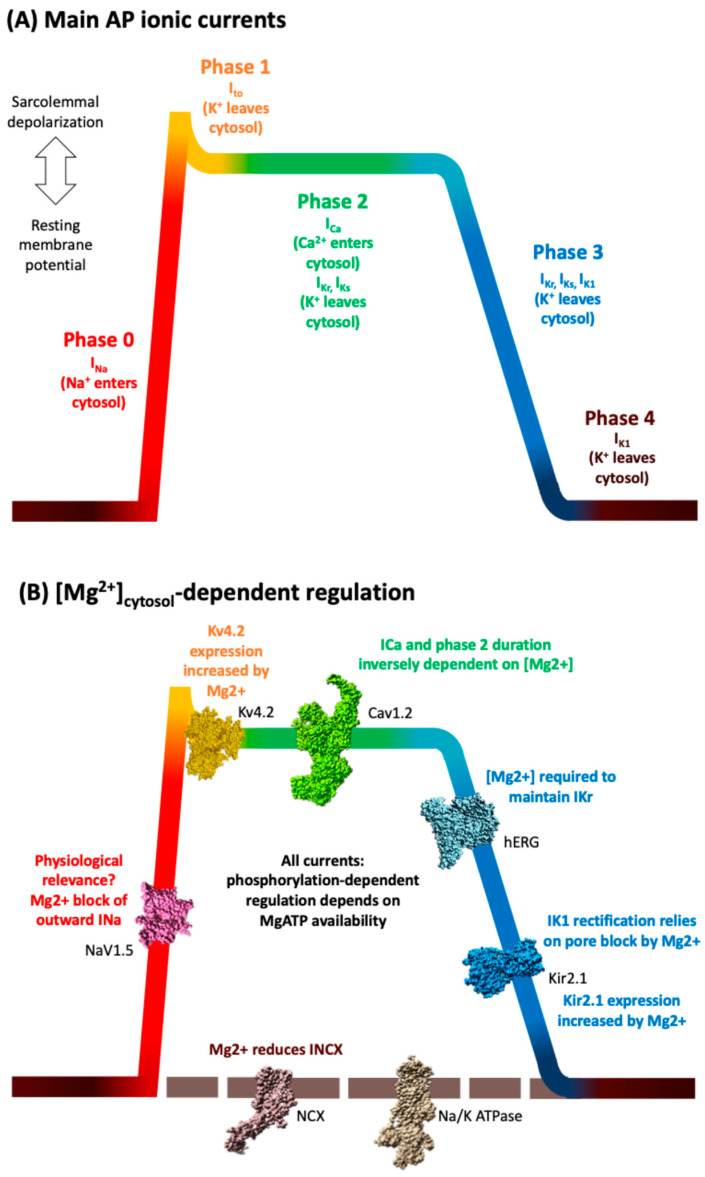
Mg^2+^-sensitive ionic currents shaping the ventricular cardiomyocyte AP. (**A**) A schematic representation of the ventricular cardiomyocyte AP phases: phase 0 (rapid depolarization), phase 1 (initial repolarization), phase 2 (plateau), phase 3 (repolarization), and phase 4 (resting potential). The top-left white arrow symbolizes the status of the membrane potential during the different phases of the AP. (**B**) Ionic channels and transporters known to be modulated by Mg^2+^ and relevant to ventricular cardiomyocyte function are indicated here. These include the rapid depolarizing sodium current INa (Nav1.5, PDB ID: 7DTD [68]), transient outward potassium current Ito (Kv4.2, PDB ID: 7F0J [69]), L-type calcium current ICaL (Cav1.2, PDB ID: 8WE6 [70]), delayed rectifier potassium currents IKr and IKs (here represented by channel hERG for IKr, PDB ID: 5VA1 [71]), inward rectifier potassium current IK1 (Kir2.1, PDB ID: 7ZDZ [72]), Na^+^/Ca^2+^ exchanger (NCX, PDB ID:8JP0 [73]), and Na^+^/K^+^ ATPase (PDB ID: 7E1Z [74]). Mg^2+^ modulates these targets through direct effects (such as pore block, voltage-dependent inhibition, or direct binding) or indirect effects (including altered channel expression, ATP availability for kinase activity), thereby influencing AP morphology, calcium cycling, and overall cardiac excitability.

## 5. The Core of CICR: Ryanodine Receptor Type 2 (RyR2)

RyR2 is the principal actor of the core mechanism of cardiac ECC, that is, CICR, a highly delicate process where even subtle defects can lead to severe cardiac pathologies, including arrhythmias and heart failure [27,75,76]. The tight coordination of events in CICR, and their extreme sensitivity to ionic variations, are supported by the precise juxtaposition in space of the protein machineries involved. Within the dyadic cleft, the principal voltage-dependent trigger, CaV1.2, lies within a dozen nanometers of the mouth of the RyR2 pore, so that small variations in Ca^2+^ abundance are rapidly perceived [25,77]. In addition, RyR2 responsiveness is also modulated from the highly Ca^2+^-enriched jSR lumenal side, where a highly specialized set of ion-sensitive interactors is located [1].

### 5.1. Mg^2+^’s Effect on the Ultrastructure of the Dyad

Within the junctional SR membrane, RyR2 clusters and superclusters form the structural basis for elementary Ca^2+^ release units that underlie ECC. These clusters exhibit distinct spatial arrangements, and Mg^2+^ concentration is a key determinant of their configuration [78,79]. Under high-Mg^2+^ conditions, RyR2 tetramers preferentially adopt a side-by-side arrangement. In contrast, low Mg^2+^ promotes a checkerboard configuration [78]. Functionally, these structural differences have profound effects: while side-by-side arrangements (favored by high Mg^2+^) are associated with lower Ca^2+^ spark frequencies, the checkerboard arrangement (favored by low Mg^2+^) leads to markedly increased spark frequencies [78].

Beyond the above effects, Mg-ATP availability determines RyR2 phosphorylation status during β-adrenergic stimulation. Such phosphorylation not only decreases the inhibitory effect of cytosolic Mg^2+^ on RyR2s [80] (see below) but also recruits PKA-phosphorylated RyR2s to the vicinity of Cav1.2s [81].

Another key determinant of dyadic ultrastructure is Junctophilin 2 (JPH2), which by binding to both sarcolemmal and jSR membranes simultaneously ensures a proper geometric relationship between both structures, and hence proper ECC. JPH2 has been shown to interact with the lipid PtdIns(3,4,5)P3, and divalent cations (Ca^2+^ and Mg^2+^) above 0.25 mM were found to perturb this association [82]. It follows that, in physiological conditions (0.5–1 mM free Mg^2+^), the presence of Mg^2+^ ensures the proper association between the jSR and T-tubule. This JHP2-mediated role of Mg^2+^ structurally safeguards the efficacy of the ECC process, which relies on the very tight ultrastructure and geometries of the dyadic space.

### 5.2. Cytosolic Modulation of RyR2

On the cytoplasmic side, RyR2 exposes a massive, 28x28 nm large platform for interaction with numerous partners and cofactors, including the Mg-ATP necessary for channel activation [27,83,84]. Here, cytosolic Mg^2+^ modulates the RyR2 open probability through multiple direct and indirect mechanisms [54,85]. Because of the high functional interdependency among cytosolic sites, and between cytoplasmic and luminal divalent-dependent processes, the precise determination of the specific properties of each Ca^2+^/Mg^2+^-binding site is an intrinsically complex challenge.

It is well known that the RyR2 response to cytosolic Ca^2+^ is biphasic: micromolar Ca^2+^ levels activate RyR2 while millimolar Ca^2+^ levels inhibit the channel. Both these processes are targets of Mg^2+^ functional effects. Mg^2+^-binding sites have been identified for skeletal RyR1 through single-particle cryo-EM experiments [84], whereas for the specific case of cardiac RyR2, no bound Mg^2+^ has been resolved. Therefore, to point out the relevant functional consequences of Mg^2+^ in cardiac ECC we will build not only on the multiple functional experimental insights into RyR2, but also upon the structural homologies and differences of Mg^2+^ binding to either cardiac or skeletal RyR.

#### 5.2.1. The A-Site

The main inhibitory effect of cytosolic Mg^2+^ on RyR2 derives from its binding to the high-affinity, activating A-site within the central domain and in proximity to the caffeine- and ATP-binding sites. The A-site mediates Ca^2+^-dependent activation with a typical Ca^2+^-affinity of approximately 1 µM, ten times the diastolic Ca^2+^ concentration [12,85,86,87,88]. Despite the 50-fold selectivity for Ca^2+^ over Mg^2+^ [86], the typical cytosolic abundance of the latter shifts the Ka of this site for Ca^2+^ up to 50 μM [26,89,90], effectively blocking spontaneous Ca^2+^ release. Mutagenesis studies on RyR2 highlighted a possible cause of the different functional outcomes of A-site binding by either Ca^2+^ or Mg^2+^. The inhibitory effect of Mg^2+^ implies binding to a set of oxygen-containing residues that, while being relevant to the Ca^2+^-activation mechanism, are not directly involved in Ca^2+^-coordination [91,92]. Recent cryo-EM analysis of the homologous RyR1 A-site revealed a mechanism that reconciles both the incomplete superposition of the Ca^2+^- and Mg^2+^-binding set of residues with their functional competition: steric fitting of a partially hydrated Mg^2+^ ion, while increasing the solvent exposure of the A-site compared to the closed state, does not cause the large cavity expansion observed in presence of Ca^2+^ (Figure 2) [84]. Mg^2+^ stabilizes an inactive yet “dormant” state, still accessible to Ca^2+^ and thus “primed” for rapid Ca^2+^-induced activation.

#### 5.2.2. The I1-Site

The inhibitory I1-site is non-selective for Ca^2+^ vs. Mg^2+^, and mediates channel closure at millimolar quantities of cytosolic divalent ions [86]. The skeletal muscle isoform features a much higher sensitivity to Mg^2+^ vs. RyR2 (with IC50 values of 0.1 vs. 2–6 mM, respectively, Table 1), which makes it very unlikely that, in physiological settings, cytosolic Mg^2+^ triggers significant RyR2 inhibition in the heart [12,27,84,85]. However, it is tempting to observe that, given the functional intertwining among multiple sites that is typical of RyRs, this minor divalent-inhibitory site may acquire a functional relevance in more pathological scenarios, such as hypermagnesemia, or during cellular Ca^2+^ overload. Because of the lack of published Mg^2+^-bound RyR2 structures, we will here rely on structural–functional studies on the skeletal isoform, RyR1, to offer interesting clues on the presumed topology of I1-sites in RyR2. Mutagenesis and structural studies on RyR1 identified two putative locations featuring the expected I1-site properties (low-affinity and non-specific divalent cation binding), where Mg^2+^ acts as a surrogate for Ca^2+^ in stabilizing the closed state [84,93]. The first putative location for the I1-site corresponds to a diffuse electrostatic surface on the junctional solenoid (Jsol) domain (Figure 2), which is a key structural lever mechanically connecting the conformational rearrangements of the cytoplasmic regulatory domains to the pore gating machinery [94]. The fact that RyR1 features a distinctive sequence of 30 negative amino acids in this region (not present in RyR2), may explain the different sensitivities to divalent cations between the two isoforms. The second putative location, the EF-hand domain, is conserved between RyR1 and RyR2, and shares the same functional characteristics. In response to the low-affinity binding of both Ca^2+^ and Mg^2+^, the EF-hand of RyR1 forms salt bridges with a neighboring subunit and stabilizes the closed conformation of the channel. As the EF-hand domains of the skeletal and cardiac isoforms share similar binding affinities, it was proposed that the higher efficiency of Cav1.1-RyR1 signal transmission in skeletal muscle ECC may functionally offset the higher RyR1 sensitivity to Mg^2+^ inhibition [93].

#### 5.2.3. The I2-Site

Differently from the I1-site, which is more functionally relevant for the skeletal RyR1, the I2-site was primarily described in RyR2. It is a cytosolic site with micromolar affinity for Ca^2+^, able to produce a partial inactivation (20–40% reduction in the open probability) in response to high levels of Ca^2+^ feed-through from the jSR [98]. While a precise identification of the residues comprising the I2-site is missing, Ca^2+^-diffusion/buffering experiments estimated that the I2-site is so much further away from the channel mouth (about 26 nm) that it may identify with an inhibitory Ca^2+^-sensing protein interactor [98]. The specific functional relevance of these I2-site-mediated inhibition processes in the context of cellular RyR2 function is not yet clear, likely because of the stronger and masking effect of Mg^2+^ inhibition at the A-site [86].

#### 5.2.4. The Channel Pore

In RyR1, robust structural data shows that Mg^2+^ binds directly within the pore (notably Asp4945, Figure 2), tightening the S6 helical bundle and increasing the resistance to pore opening [84]. MD experiments confirmed that this coordination, occurring exclusively in the closed state, is selective for Mg^2+^ over Ca^2+^ [93]. Since these structural insights on Mg^2+^-coordination at the pore functionally belong to the low-affinity cytosolic Mg^2+^ inhibition mechanism in RyR1, and considering the high sequence homology between RyR2 and RyR1 in the pore-forming region [107], it is reasonable to infer that a comparable direct pore occlusion mechanism could exist in RyR2 yet be far weaker. What is known with certainty is that RyR2 is permeable to Mg^2+^ just as it is to Ca^2+^, so that, under resting conditions, the expectedly symmetric distribution of Mg^2+^ across the SR membrane renders RyR2 a functionally Mg^2+^-selective pore in terms of occupancy. Even though net current flow remains Ca^2+^-selective, the presence of Mg^2+^ halves it [54]. In addition, the presence of Mg^2+^ in the pore directly points to the so-called “feed-though” effect of luminal Mg^2+^, theoretically able to reach the cytosolic A-site and thus inhibit RyR2. However, as the variation in Mg^2+^ may be lower than 1 mM, the extent of such an effect may be strongly limited [27]. In failing hearts or catecholaminergic polymorphic ventricular tachycardia (CPVT), RyR2 sensitivity to Mg^2+^ inhibition is decreased, promoting arrhythmogenic Ca^2+^ release [76,83].

#### 5.2.5. The ATP-Binding Pocket

In addition to the aforementioned divalent cation binding sites, it has to be considered that cytosolic Mg^2+^ also permits the binding of ATP, in the Mg-ATP complex form, within the cavity between central domains and the C-terminal domain (Figure 2). However, the different allosteric dynamics of RyR1 and RyR2 lead to distinct outcomes: while the presence of Mg-ATP is dispensable in RyR1 for Ca^2+^-independent channel opening, in RyR2 it rather induces a “primed” state, making the channel more susceptible to Ca^2+^-induced activation [83]. ATP activation of both RyR1 and RyR2 has an EC50 in the range of 0.3–1 mM (Table 1). Given the physiological abundance of 5 mM Mg-ATP in cardiac cells, its maximal effect on RyR1/2 is always exerted [2]. Of note, it has been shown that presence of Mg-ATP on the cytosolic side of RyR2 is essential for the luminal partner CASQ2 to inhibit the RyR2 channel [108].

#### 5.2.6. The Ca^2+^/Mg^2+^-Sensitive Cytosolic Interactome of RyR2

The uniquely large cytosolic portion of the RyR2 tetramer also allows for interactions with a wealth of protein partners and enzymes. Among them, Calmodulin (CaM) is a key and ubiquitous Ca^2+^ sensor that modulates RyR2 directly through controlled activation of RyR2 kinases. Mg^2+^ can reduce the apparent Ca^2+^ affinity of CaM, thus affecting its ability to bind to and regulate its multiple targets [20,109].

Among the numerous kinases activated by Ca^2+^-bound CaM, the delta isoform of the Ca^2+^/calmodulin-dependent protein kinase II (CaMK2d) is a critical modulator of cardiac RyR2. Physiologically, CaMK2d permits ventricular myocytes to adapt its AP and ECC machinery to increases in heart rate. However, when pathologically hyperactive, CaMK2d is involved in the pro-arrhythmogenic effects of catecholaminergic stimulation in the heart, in part mediated by phosphorylation of the target RyR2 residues Ser2814 (Ser2815 in large mammals) and Ser2808 (Ser2809 in the tested canine model), which enhances RyR2-mediated diastolic Ca^2+^ release [110,111]. In resting myocytes, most CaMK2d is found near RyR2s. Upon pacing and binding of Ca^2+^/CaM to CaMK2d, a portion of the kinase diffuses away from Ca^2+^ sources and longitudinally into the sarcomere, thus reaching a wider number of targets [112,113,114].

Here, the cycling variations in local Ca^2+^ concentrations allow for transient binding of Mg^2+^ at Ca^2+^- binding sites on CaMK2d, ultimately inhibiting the enzyme activity. Regarding instead a more general cytosolic mechanism, low Mg^2+^ levels promote the phosphorylation of target Na^+^ and Ca^2+^ channels, including RyR2 [115,116].

In addition to CaMK2d, the cAMP-dependent protein kinase A (PKA) also necessitates Mg-ATP to efficiently phosphorylate RyR2 [117]. The crystallographic PKA–RyR2 peptide complex (PDB ID: 6MM5), where Mg^2+^ coordinates the ATP at the PKA active site. is the only Mg^2+^-containing published structure where RyR2 appears. Similarly to Camk2d, PKA activity on multiple serines of RyR2 leads to uncoupling of the channel open probability from cytosolic Ca^2+^ inactivation, which in turn facilitates SR calcium leak [118,119,120]. In mice, the only two sites where PKA-dependent RyR2 phosphorylation have been found to be of physiological relevance are S2808 (S2809 in large mammals) and S2030 [76,83,121,122].

Crucially, β-adrenergic stimulation, which increases RyR2 phosphorylation by CaMK2d and PKA at key sites such as S2808, S2814, and S2030, significantly diminishes the inhibitory effects of both luminal and cytoplasmic Mg^2+^ on RyR2 activity [8,79,95]. This phosphorylation-dependent reduction in Mg^2+^ inhibition is particularly evident in heart failure, where RyR2 hyperphosphorylation correlates with decreased channel sensitivity to cytoplasmic Mg^2+^ [95]. In addition, it has been shown that a sizable Mg^2+^ extrusion occurs in ventricular myocytes within minutes of β-adrenergic stimulation [123], which could further contribute to the β-adrenergic-stimulation-dependent increase in the apparent RyR2 affinity for Ca^2+^ in intact cells.

### 5.3. Luminal Modulation of RyR2

It is unequivocally established that luminal Mg^2+^ is a potent inhibitor of cardiac RyR2 activity. Mechanistically, two pathways have been proposed: Mg^2+^ may flow through the channel to bind the cytosolic A-site, thereby preventing further channel activation by Ca^2+^, or bind directly to luminal sites to inhibit gating [86]. However, multiple studies indicate that the magnitude of the former “feed-through” mechanism is unlikely to be an efficient driver of the store–load dependence of RyR2 opening in vivo [27,86]. Experimental evidence, primarily obtained using RyR2 samples preserving associated luminal proteins such as calsequestrin (CASQ2), triadin (TRDN), and junctin (JNT), unveiled the role of the so-called L-site, an activation site for luminal Ca^2+^, at which luminal Mg^2+^ competes as a non-activating antagonist [12,100]. Here, luminal Mg^2+^ reduces the sensitivity of RyR2 to cytosolic Ca^2+^ and caffeine-mediated activation, requiring higher concentrations of these cytosolic activators to achieve channel opening [27]. This competition shifts the Ca^2+^ dose–response toward higher luminal Ca^2+^ and serves as a critical physiological brake to prevent diastolic SR Ca^2+^ leak [12,86]. Of particular note, unlike cytosolic Mg^2+^ inhibition (which is dominant in RyR1 but significantly weaker in RyR2), luminal Mg^2+^ exerts a more specific and potent inhibitory effect on RyR2. This highlights the importance, in the cardiac setting, of those Mg^2+^-dependent mechanisms specifically occurring within the jSR lumen.

In the following paragraphs we will delve into how luminal Mg^2+^ regulates the open probability of RyR2 either in presence or absence of its soluble accessory protein CASQ2. Intriguingly, these two mechanisms, featuring distinct selectivities for Ca^2+^ vs. Mg^2+^ ions and different magnitudes of their physiological impacts, also feature opposite functional outcomes on RyR2.

**Figure 2 cells-14-01280-f002:**
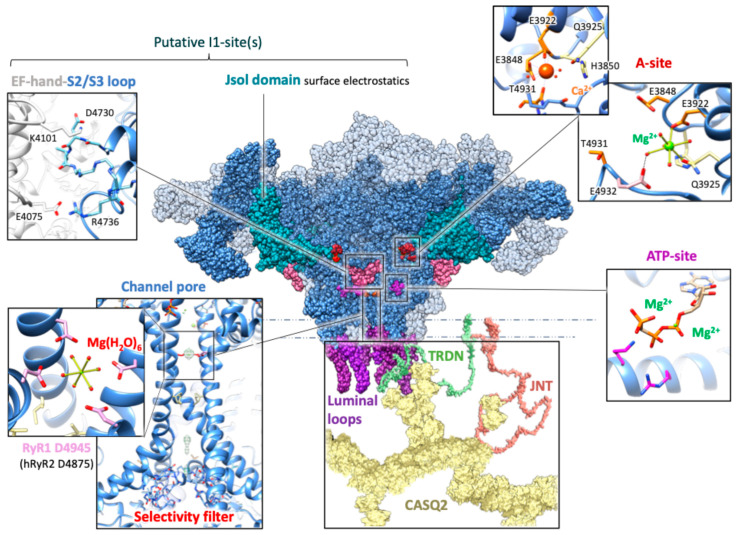
The structural landscape of Ca^2+^/Mg^2+^ binding sites in RyR2. The central panel shows the human cardiac ryanodine receptor type 2 (RyR2) structure (PDB ID: 7UA5 [124]) in its closed conformation, with two of the four monomers displayed as semi-transparent to enhance visualization of internal domains. The **A-site** (highlighted in red on the RyR2 structure): Distinct binding mechanisms are shown in the relative panels. In the open RyR2 structure (PDB ID: 7UA9 [124]), cytoplasmic Ca^2+^ is directly coordinated by the carboxylate side chains of Glu3848 (E3848) and Glu3922 (E3922) from the core domain and the backbone carbonyl of Thr4931 (T4931) from the C-terminal domain, with Gln3925 (Q3925) and His3850 (H3850) contributing to the second coordination sphere. Ca^2+^ binding promotes separation between the central and C-terminal domains, stabilizing the highly active open state. In contrast, Mg^2+^ binding has been experimentally visualized only in RyR1 (PDB ID: 7UMZ [84]). Given the conservation of residues, residue numbering here refers to human RyR2. In RyR1, Mg^2+^ coordinates predominantly with Glu3967 and Gln3970 (corresponding to RyR2 Glu3922 and Gln3925), and indirectly with Glu5002 (RyR2 Glu4932) via a water molecule, remaining separated from Glu3893 (RyR2 Glu3848). This distinct Mg^2+^ coordination results in minimal conformational changes, insufficient to trigger channel activation. The **I1-site** inhibitory mechanism expectedly involves either one or both of two domains: the EF-hand/S2-S3 loop interface (pink), stabilized by salt bridges in the Mg^2+^-triggered closed RyR1 conformation (PDB ID: 7UMZ [84]), and the Jsol (handle) domain (teal blue), with a highly charged surface favorable for divalent cation binding. The **ATP-binding site** (magenta) is detailed in the corresponding panel, showing two Mg^2+^ ions coordinating ATP as resolved in RyR1 under high-Mg^2+^ conditions (PDB ID: 7UMZ [84]). **The channel pore:** In RyR1 (PDB ID: 7UMZ [84]), a hydrated Mg^2+^ ion is identified, positioned to form an octahedral water shell coordinated by D4945 from each monomer (corresponding to D4875 in human RyR2), stabilizing the closed state. Nearby lies the **selectivity filter**, another conserved charged region where Ca^2+^ and Mg^2+^ may compete, as supported by small non-protein densities observed in the RyR1 cryo-EM map (EMD-26610 [84]). **Luminal loops** (purple) are shown, where intrinsic luminal Mg^2+^-dependent activation occurs. The disordered first intraluminal loop (RyR2 residues 4524–4555), structurally unsolved, is graphically represented as an extended loop. Both intraluminal loops harbor potential interaction sites, with **CASQ2** (represented using PDB ID 6OWW [125]) **and TRDN** mediating voltage-independent, luminal-Mg^2+^-dependent inactivation. The functional outcome of RyR2 physiological interaction with JNT is less understood.

#### 5.3.1. Mg^2+^ Direct Binding at RyR2 Luminal Site(s) Cooperates with Ca^2+^ Activation

When RyR2 is stripped of CASQ2, maximally activating concentrations of cytosolic Ca^2+^ (100 µM) unmask an intrinsic luminal Mg^2+^-activation mechanism, where luminal Mg^2+^ is equally efficient as luminal Ca^2+^ in sustaining RyR2 channel opening, with an EC50 for Ca^2+^/Mg^2+^ activation of 379 ± 247 µM [27,103]. More recently, Magyar et al., by probing at diastolic cytosolic Ca^2+^ (100 nM) levels with a non-permeating ion (Eu^3+^), corroborated the existence of an intrinsic luminal Ca^2+^-activating site in canine RyR2 [100].

It is interesting to observe here that, differently from RyR2, the genuine luminal Ca^2+^-binding site of the skeletal channel has an inhibitory effect [100]. In addition, while the Ca^2+^ affinity of the RyR2 L-site does not depend on voltage, the effect of cation binding at the RyR1 luminal site is voltage-dependent. Because of this non-conserved mechanism, two distinct locations have been proposed for the intrinsic L-sites of both the cardiac and skeletal channel isoforms. The luminal Ca^2+^/Mg^2+^-dependent inhibition of RyR2 is proposed to be mediated by a region outside the membrane’s electric field. The EF-hand pattern found specifically within RyR2’s first intraluminal loop (connecting the transmembrane helices S1 and S2) features this requirement and the absence of divalent cation selectivity [100,101]. On the contrary, the voltage-sensitive RyR1 inhibitory luminal site appears to be situated within the channel pore. On these premises, the intrinsic L-site of the skeletal channel has been proposed to involve the selectivity filter: a negatively charged, conserved GGGIG motif (lying within the S5–S6 luminal loop) exposed at the luminal mouth of the channel pore [100].

Recent studies also unveiled an additional, intrinsic, RyR2 luminal Ca^2+^-binding site or Ca^2+^-dependent allosteric mechanism which becomes apparent only under particular cytosolic Ca^2+^ concentrations and is otherwise of questionable functional importance [100]. The location of this site has been proposed to be conserved between isoforms: close to the channel gate and the selectivity filter. However, there are currently neither functional nor structural data on the effect here played by Mg^2+^: while cryo-EM analysis of RyR1 shows an axial non-protein density in the conserved selectivity filter region, its density did not increase with Mg^2+^ concentration [84].

#### 5.3.2. RyR2 Inhibition by Luminal Mg^2+^ Is Mediated by IDR-Containing Proteins

Since the effects of luminal Mg^2+^ on RyR2 in the absence of protein partners are not specific to Ca^2+^ or Mg^2+^, and their physiological meaning is only evident in special circumstances, the presence of luminal protein partners is the missing link to mechanistically explain the well-known, physiologically relevant RyR2 inhibition by luminal Mg^2+^ (Figure 2). Partners ensure that RyR2 is capable of discriminating between luminal Ca^2+^ (which activates RyR2 when interacting with partners) and luminal Mg^2+^ (crucial to prevent excessive Ca^2+^ release, particularly during diastole when cytoplasmic Ca^2+^ concentrations are low and the SR is well loaded with Ca^2+^) [12,86,98]. Furthermore, in a context where the physiological concentration of luminal Mg^2+^ is considered to stably remain at approximately 1mM (due to the absence of well-established active Mg^2+^ transport across the SR membrane), the discrimination between divalent cations is also necessary to ensure that the RyR2 activity is coupled with physiological fluctuations in the luminal free Ca^2+^ content between 0.3 and 1 mM [3,86]. In addition, while partner-independent mechanisms alter the maximal efficacy of cytosolic Ca^2+^-dependent activation of RyR2, the CASQ2-dependent mechanism modulates the sensitivity to cytosolic Ca^2+^ [103].

Crucially, only in presence of all its physiological protein partners (TRDN, JNT, and CASQ2), RyR2 is sensitive to both luminal Ca^2+^ and Mg^2+^, with comparable affinities of 35–45 µM [126]. Mechanistically, competing luminal Mg^2+^ shifts the K_a_ for luminal Ca^2+^ activation to 1 mM, close to the physiological variations in luminal Ca^2+^.

Notably, all members of the RyR2 macromolecular complex interact through their intrinsically disordered regions (IDRs) within the jSR lumen, with the globular core of CASQ2 representing a structured exception. Of note is the fact that the extended and structurally unresolved first luminal loop of RyR2 (connecting transmembrane helices S1–S2) may also contribute to this unstructured signaling hotspot [100,127]. The fascinating biochemical nature of the intricate interactions between RyR2 and its physiological partners is dissected below.

#### 5.3.3. CASQ2

While free luminal Ca^2+^ cycles in the 0.3–1 mM range, Ca^2+^ ions in the jSR actually total 19 mM [1]. This impressive storage capacity builds upon the multiple low-affinity Ca^2+^-binding sites of a highly acidic protein, namely CASQ2. The significant proportion of aspartate and glutamate residues, of which half are concentrated in its C-terminal tail, endows CASQ2 with the ability to bind up to 60 ions per 45 kDa monomer while assembling into its functionally relevant, polymeric form [128,129]. CASQ2 is tethered to its jSR transmembrane partners TRDN and JNT [130,131], and inhibits RyR2 channel activity, with maximal inhibition at 1 mM Ca^2+^. In line with the physiological trend of Mg^2+^ competition for Ca^2+^-binding sites, Mg^2+^ reduces the number of Ca^2+^ ions bound to CASQ2 [131,132], implying that Mg^2+^ could indirectly impact the functionality of CASQ2 within the RyR2 complex and ultimately affect jSR Ca^2+^ buffering and release dynamics [133]. Notably, Mg^2+^ has been shown to “prime”, by an unknown molecular mechanism, recombinant CASQ2 for efficient Ca^2+^-dependent polymerization. Collectively analyzed, published insights demonstrating the dominant role of CASQ2’s disordered C-terminal portion on the protein polymerization responsiveness to Ca^2+^ [134], and on its interaction with RyR2 [135], can be interpreted as indicating that the main functional outcomes of Ca^2+^/Mg^2+^ competition are determined at this domain of the protein. This said, new studies are needed to further strengthen this interpretation.

Importantly, and unlike the RyR2-intrinsic L-site, the CASQ2-mediated regulation mechanism of RyR2 distinguishes between luminal Ca^2+^ and Mg^2+^, as variations in luminal Ca^2+^, but not in luminal Mg^2+^, alter the cytosolic Ca^2+^ sensitivity of the channel [103]. There is no clear consensus regarding the specific site of binding between CASQ2 and RyR2, the possible involvement of direct Mg^2+^ binding, or its proximity to the intrinsic L-site (whose activating effect is masked by CASQ2-dependent inhibition) [104,120,130,136,137]. The prevailing understanding is that CASQ2 modulates RyR2 activity in a Ca^2+^-dependent manner through its physical interactions with TRDN [35,103,138].

#### 5.3.4. TRDN and JNT

TRDN and JNT are single-pass transmembrane proteins which homologous, extended, and highly flexible luminal tails both possess multiple KEKE (Lys-Glu-Lys-Glu) motifs, believed to be the primary binding sites for Ca^2+^, RyR2, and CASQ2 [139,140]. They are not only essential for anchoring CASQ2 to the jSR membrane, but also for luminal Ca^2+^ sensing and propagation of conformational changes across the RyR2 protein complex [136,141,142,143]. On the RyR2 side, the site for interaction with TRD and JNT is presumed to lie within the second intraluminal loop, particularly conserved and rich in charged amino acids [35]. However, in RyR1 it has been demonstrated that the C-terminal loop is also essential [105]. Notably, while the interaction between TRDN and JNT with RyR2 is Ca^2+^-independent [35], their interaction with CASQ2 is inhibited by increasing Ca^2+^ amounts [137]. Of note, while the role of cardiac TRDN in physically tethering CASQ2 to the RyR2 complex is solidly demonstrated [134], the role of JNT is far less evident. One study even reported that JNT does not directly interact with CASQ2 [144]. What is known regarding the role of JNT in the cardiac setting is that it indirectly supports proper CASQ2 anchoring at the jSR membrane, and that its overexpression leads to narrowing and extension of the jSR portion [145]. Because of its poor relevance in the cardiac myocytes (counteracted by a more conspicuous effect in the skeletal scenario), we will not dissect its properties further.

#### 5.3.5. The Functional Relevance of Luminal IDRs in the RyR2 Complex

Either TRDN or CASQ2 knockout leads to severe architectural disorganization of jSR cisternae, and a notable reduction in the levels of the counterpart protein: CASQ2 in the case of TRDN KO, and TRDN in the case of CASQ2 KO. Both these situations lead to impaired Ca^2+^ homeostasis and enhanced RyR2 sensitivity to cytosolic Ca^2+^ activation [108,136,146,147,148], which in turn couples with the development of CPVT, a highly severe hereditary arrhythmogenic disorder [103,149,150,151].

All this evidence for the profound functional link between TRDN and CASQ2 points to the fact that there exists at least a partial superposition among their mechanisms of interaction with RyR2. In the case of TRDN, the molecular basis of this functional link can only be attributed to its long and disordered luminal segment. Interestingly, concerning the mechanism employed by CASQ2 for Ca^2+^-dependent RyR2 modulation, what is known is that the electrostatics of its C-terminal IDR integrates the sensitivity to multiple ions (Ca^2+^ and Mg^2+^) within the dynamics of the functional interactions between CASQ2 monomers [125,133] and expectedly between CASQ2 and the TRDN-RyR2 complex [134,140]. Further supporting the relevance of the C-terminal IDR and of its multiple dynamic interactions with the globular core, CPVT-causative missense CASQ2 mutations are spread all over the negative surface of CASQ2, with the specific exception of this C-terminal segment [130]. Overall, it appears that the highly electrostatic IDRs exposed by CASQ2 and TRDN funnel the multiple Ca^2+^/Mg^2+^-competitive binding events into a defined set of (expectedly similar) mechanisms for direct and/or indirect RyR2 modulation.

## 6. Pathological Dysregulation of Mg^2+^ and Therapeutic Implications

The tightly controlled and delicate balance of intracellular free Mg^2+^ is critical, and deviations in either direction (hypomagnesemia or hypermagnesemia) have profound pathological implications. Mg^2+^ is one of the few ions used as a first-line treatment in emergency cardiovascular care: intravenous magnesium sulfate is administered for Torsades de pointes and other ventricular arrhythmias, and shows potential in long-term heart failure and hypertension management and as a metabolic stabilizer in diabetes [10].

### 6.1. Hypomagnesemia and Associated Disorders

Hypomagnesemia (commonly defined as serum Mg^2+^ < 0.8 mM) is increasingly recognized as a widespread and underdiagnosed condition [152]. Given that serum Mg^2+^ represents only ~1% of total body magnesium, patients can be in a Mg^2+^-depleted state even with normal serum levels and remain asymptomatic until serum Mg^2+^ falls below 0.5 mM, highlighting why the clinical impact of magnesium deficiency may be underestimated [152]. Signs of hypomagnesemia include generalized weakness to cardiac ischemia and death. Hypomagnesemia is also prevalent in metabolic (diabetes and obesity) and neuromuscular disorders [10,153,154]. In the skeletal muscle, pathological RyR regulation and oxidative stress are exacerbated by Mg^2+^ deficiency [155]. Also, since Mg^2+^ exerts a vasodilatory effect on vascular smooth muscle, its deficiency promotes vasoconstriction and hypertension [48]. The number of consequences of hypomagnesemia are due to the multiple pathways affected by Mg^2+^ within each tissue, which could secondarily affect heart function.

More specifically for the heart, hypomagnesemia alters not only the contractile force of the actin–myosin system, but also the handling of the electrical signal and ECC (Figure 3). The overall outcome on cardiac health mainly falls into two conditions:Heart Failure (HF): Chronic heart failure is often associated with reduced intracellular free Mg^2+^. In a canine pacing-induced HF model, [Mg^2+^]i was reduced by ~50% in failing cardiomyocytes [156], leading to enhanced Ca^2+^ influx, slower inactivation, and increased susceptibility to Ca^2+^ overload, hypertrophy, and arrhythmias [52]. Beyond Ca^2+^ handling, Mg^2+^ deficiency alters action potential dynamics. Depending on cell type and repolarization mechanisms, low [Mg^2+^]i levels can prolong AP duration by inhibiting K^+^ or prolonging Ca^2+^ currents [7,10]. These effects facilitate early (EADs) and delayed (DADs) afterdepolarizations, especially under β-adrenergic stimulation. Disease progression in HF is also due to the negative consequences of low Mg^2+^ on mitochondria (oxidative stress, inflammation, and metabolic dysfunction), partly through upregulation of the TRPM7 kinase-channel [5].Arrhythmias: Mg^2+^ deficiency prolongs AP duration by affecting K^+^ and Ca^2+^ currents and, as shown in Section 5, increases RyR2 diastolic leak (Figure 3) [1]. These changes increase susceptibility to EADs and DADs, Torsades de pointes [60], atrial fibrillation and CPVT [48]. In CPVT, defective regulation of RyR2—often due to mutations in calsequestrin (CASQ2)—is compounded by low Mg^2+^ levels, worsening the arrhythmic phenotype [157].

**Figure 3 cells-14-01280-f003:**
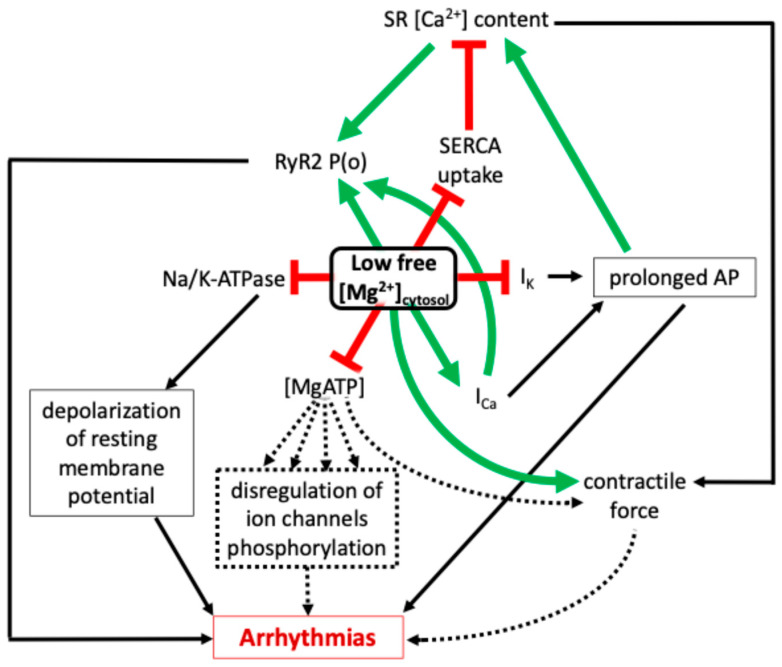
Integrated effects of low cytosolic free magnesium concentration ([Mg^2+^]_cytosol_) on cardiac ECC and arrhythmogenesis. The green arrows indicate positive (enhancing) effects; red bars indicate inhibitory effects; and dotted arrows represent indirect regulatory pathways for which only the net positive/negative effect has not been precisely determined. This multifaceted disturbance underscores the essential role of Mg^2+^ homeostasis in maintaining cardiac electrical and contractile stability. Low intracellular free Mg^2+^ profoundly alters cardiac physiology at multiple levels. At the membrane level, it enhances L-type Ca^2+^ current (I_Ca_) and suppresses repolarizing K^+^ currents (I_K_), contributing to prolonged action potential duration (APD). APD prolongation facilitates increased Ca^2+^ influx and sarcoplasmic reticulum (SR) Ca^2+^ content. In parallel, low Mg^2+^ enhances RyR2 open probability (P_o_) by relieving Mg^2+^-dependent inhibition at both cytosolic and luminal Ca^2+^ regulatory sites, further promoting SR Ca^2+^ leak. Low [Mg^2+^]_cytosol_ also reduces the pool of usable MgATP, which is essential for the activity of several ATPases. Decreased MgATP impairs SERCA-mediated Ca^2+^ reuptake into the SR and diminishes Na^+^/K^+^-ATPase activity, leading to membrane depolarization and destabilization of ionic gradients. MgATP scarcity also disrupts kinase/phosphatase activity and ion channel phosphorylation states, compounding electrical instability. In the myofilament compartment, reduced Mg^2+^ increases the Ca^2+^ sensitivity of troponin C by decreasing competitive inhibition, potentially enhancing contractile force. However, impaired Ca^2+^ reuptake due to SERCA inhibition and RyR2 leak can deplete SR Ca^2+^ reserves over time, compromising systolic Ca^2+^ transients and contractile efficiency. The net result is a paradoxical state where increased Ca^2+^ release coexists with impaired relaxation and energy imbalance, predisposing to arrhythmias.

### 6.2. Ischemia-Associated Hypermagnesemia

Although clinically less common, during transient ischemia, intracellular Mg^2+^ rises two- to three-fold due to ATP hydrolysis [158], leading to potent inhibition of L-type Ca^2+^ currents and accelerated inactivation [52]. This Mg^2+^-mediated inhibition of Ca_V_1.2, coupled with depressed myofilament Ca^2+^ sensitivity and ATPase activity, serves as a cell-autonomous protective mechanism, reducing Ca^2+^ influx, contractile demand, and metabolic burden in affected myocytes while sparing neighboring healthy cells [159].

## 7. Conclusions

Endowed with a high charge density and a massive solvation sphere, Mg^2+^ ions have difficulties in stably binding to flexible protein binding sites. Coordination with Mg^2+^ is rigid and energetically costly, making Mg^2+^ unsuitable for signaling functions such as cardiac ECC. This signaling role is ultimately performed by Ca^2+^, a more flexible bivalent with lesser desolvation requirements and a promiscuous coordination, which allows proteins to flexibly change conformation upon ion (un-)binding. That said, Mg^2+^ exerts surprisingly strong effects on cardiac ECC, either in the form of direct regulation (e.g., IK1 rectification, RyR2 inhibition by competition with Ca^2+^) or much subtler pathways, such as alteration of protein distribution (e.g., changes in JPH2 conformation and RyR2 disposition), control of protein expression (e.g., Kir4.2 and 2.1), and indirectly via Mg-ATP availability, which impacts resting membrane potential, RyR2 proneness to open (via ATP site), and overall channel phosphorylation. Three thrilling frontiers to be researched further are the elucidation of the Mg^2+^-dependent interaction with SR luminal disordered regions (such as those within CASQ2 and TRDN); structural determinants of Mg^2+^ interactions with RyR2; and how such Mg^2+^-dependent interactions play a pro-arrhythmogenic role in diseased states. While the leading role of Ca^2+^ in cardiac ECC cannot be overstated, the quiet and steady antagonistic action of Mg^2+^ provides what cardiac cells need most: robustness and stability.

## Figures and Tables

**Table 1 cells-14-01280-t001:** A summary of known Ca^2+^ and Mg^2+^ binding sites in ryanodine receptor isoforms. The table lists binding sites for Ca^2+^ and Mg^2+^ identified in ryanodine receptor type 1 (RyR1, skeletal muscle) and type 2 (RyR2, cardiac muscle). For each site, the structural location, known coordinating residues, relative affinity for Mg^2+^ versus Ca^2+^, and functional outcomes of Mg^2+^ binding are summarized. Sites include the high-affinity cytosolic activation site (A-site), inhibitory sites (I1 and I2), the EF-hand/S2-S3 loop interface, the Jsol (handle) domain, ATP-binding sites, the selectivity filter, and luminal sites. Where available, structural data supporting Mg^2+^ coordination are indicated. For RyR2, Mg^2+^ binding has often been inferred based on homologous sites resolved in RyR1 structures. References on the structural location of the functional sites have been categorized as follows: ^a^: based on homology with RyR1, for which structural evidence is reported; ^b^: based on mutagenesis or trypsinization studies; ^c^: based on rational or computational hypotheses. P_o_: open probability.

Side	Site	Protein Location	Ion Binding Properties	Functional Consequence of Mg^2+^
**Cytosolic**	**A-site**	Within central domain, in proximity to C-terminal domain and to ATP-binding site [84] ^a^, [91] ^b^	K_a_ for Ca^2+^= 2–5 µM K_a_ for Mg^2+^= 50 μM [12,86,93,95,96]	Mg^2+^ shifts the apparent K_a_ for Ca^2+^ up to 50 μM [86] and stabilizes an inactive state yet accessible to Ca^2+^ [84].
	**I1-site**	Two putative locations: Electrostatic surface of JSol domain [97] ^c^ Interface between EF-hand and S2-S3 loop [84] ^a^, [93] ^c^	RyR2 IC_50_ Mg^2+^= 2–10 mM RyR1 IC_50_ Mg^2+^= 0.1 mM [12,27,84,85]	Mg^2+^ acts as a surrogate for Ca^2+^ in stabilizing the closed state. Physiological cytosolic Mg^2+^ is unlikely to trigger an inhibitory mechanism through the I1-site in cardiac muscle.
	**I2-site**	Estimated at a 26 nm distance from the cytosolic channel mouth; possibly identifies with an inhibitory, Ca^2+^-sensing protein interactor [98]	RyR2 K_a_ for Ca^2+^ = 1 µM [98]	Mg^2+^ produces partial inactivation (20–40% reduction in open probability) in response to high levels of Ca^2+^ “feed-through” from the jSR. Physiological Mg^2+^ is largely unlikely to impact RyR2.
	**ATP-Binding Pocket**	Cavity between the central domains (U-motif and S6) and the C-terminal domain [84] ^a^	EC_50_ for Mg-ATP = 0.2 mM [99]	Binding of Mg-ATP induces a “primed” state, more susceptible to Ca^2+^-induced activation. Mg-ATP is essential for the luminal partner CASQ2 to inhibit RyR2 channel activity.
**Transmembrane**	**Pore**	In RyR1: D4945 (corresponding to D4875 in RyR2) lining the pore in S6 helices [54] ^c^, [84] ^a^	RyR1: Selective for stable Mg^2+^ binding over Ca^2+^ [84]	RyR1: Mg^2+^ stabilizes the closed state, increasing resistance to pore opening [84]. RyR2: Mg^2+^ at this site is not described. The RyR2 channel is permeable to Mg^2+^ just as it is to Ca^2+^ [54].
	**Selectivity** **Filter**	Conserved GGGIG motif within the S5–S6 loop (residues 4790–4830 in human RyR2) [54,84]^c^	RyR1 and RyR2: Higher charge density expectedly favors Mg^2+^ binding over Ca^2+^ [54] RyR1 IC_50_ Eu^3+^ = 0.4 mM at 100 nM [Ca^2+^]_cyt_ [100].	There is expected similarity with the biphasic effect of Ca^2+^ and Eu^3+^ on RyR2, activating at submicromolar concentrations and inhibiting at higher than 1 µM concentrations [100].
**Luminal**	**Intrinsic Luminal** **Activation site**	Proposed within intraluminal S1-S2 EF-hand motif [98] ^b^, [100,101] ^c^, and proximal to S6-helical bundle [102] ^b^	RyR2 EC_50_ Ca^2+^/Mg^2+^ = 0.2–0.5 mM [103]	During pathological SR overload, when the inhibitory interaction with partners is lowered, maximal open probability is enhanced by luminal Ca^2+^/Mg^2+^ ions.
	**Intrinsic Luminal** **Inhibition site**	Proposed close to channel activation gate, near RyR1 Q4933 [100,101]^c^	RyR2 P_o_ inhibited by 15 µM luminal Eu^3+^ at 100 nM [Ca^2+^]_cyt_ [100]	Unknown.
	**CASQ2-** **dependent** **Inhibitory** **L-site**	Proposed binding of CASQ2 at RyR2-specific S1-S2 EF-hand [104]^b^ Proposed binding of TRDN at S3-S4 loop [105,106] ^b^	Voltage-independent K_a_ for Ca^2+^/Mg^2+^ = 35–45 µM at 100 μM luminal Ca^2+^ [98] Hill coefficient = 2	At physiological concentrations, luminal Mg^2+^ shifts the Ka for luminal Ca^2+^ activation to 1 mM.

## Data Availability

No new data were created or analyzed in this study. Data sharing is not applicable to this article.

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
