# Peer review of "The Yin and Yang of Heartbeats: Magnesium–Calcium Antagonism Is Essential for Cardiac Excitation–Contraction Coupling"

_cells, 2025, doi:10.3390/cells14161280_

Round 1
Reviewer 1 Report
Comments and Suggestions for Authors
The present review by Marabelli et al., entitled “The Yin and Yang of Heartbeats: Magnesium-Calcium Antagonism is Essential for Cardiac Excitation–Contraction Coupling”, focuses on the roles played by magnesium ions (Mg2+) in cardiac excitation-contraction coupling process. The review nicely summarizes different steps of EC-coupling magnesium competes with calcium in fine tune regulation of action potential generation, calcium-induced calcium release, sarcoplasmic reticulum regulation, modulation of contractile myofilaments, and calcium extrusion during relaxation. The manuscript provides a detailed and comprehensive description of structural, biophysical, molecular, and electrophysiological mechanisms involved in magnesium-calcium antagonism. This review is timely; the manuscript is well-written and provides an objective overview of current literature on this topic. I just have a few minor suggestions and concerns listed below.
- It would be beneficial for readers to add one sentence at the end of the abstract concluding the main purpose/scope of the present review as it will appear on PubMed.
- The review would benefit from including a summary schematic combining all the key effects of magnesium on EC-coupling steps. This can be shown as a schematic or using a classic cartoon by Donald Bers representing spatial arrangement of EC-coupling and contraction steps in a micro-domain specific manner.
- It might be reasonable to add a separate section at the end of the manuscript that will summarize the role of magnesium in the regulation of EC-coupling and, potentially, cardiac arrhythmias in the pathological settings, such as heart failure.
- Similarly, it will be beneficial to add a brief section highlighting clinical relevance and potential therapeutic potential of magnesium in treating abnormal cardiac contraction/altered EC-coupling and, potentially, cardiac arrhythmias, such as torsades de pointes where magnesium is considered as the first-line treatment option.
- The authors should consistently use “AP” as an abbreviation for “action potential” once it is introduced at the beginning of the manuscript. The same is true for all other abbreviations. For example: Line 240 – There is no need to re-introduce ECC again.
- Lines 284-287: In Ref. 60, the authors showed that “decreases in the intracellular Mg(2+) concentration, to levels outside their normal ranges prolonged action potential duration by decreasing the I(K1) transient.” Indeed, clinically, hypomagnesemia is associated with torsade de pointes which relies on APD prolongation and development of EADs. Hin contrast, as currently stated in line 287, APD shortening would rather suppress EAD formation. Please, check this.
- Line 385: “RyR21” – should be “RyR1”.
- Line 487: What is about another CaMKII-dependent phosphorylation side 2808?
Author Response
We thank both reviewers for their thoughtful and constructive comments, which greatly contributed to improve the quality and clarity of the manuscript. We appreciate the positive evaluation of the relevance and timeliness of this review study on the moonlighting role of Mg2+ in cardiac function. Following the insightful suggestions for refinement, we included now a schematic summary (Figure 3) to visually integrate the effects of Mg2+ on ECC components, and a new clinically-focused section summarizing the key pathological implications and therapeutic potential of magnesium in cardiac diseases. We have made several revisions to key paragraphs and Table 1 to improve precision and acknowledge recent findings, particularly regarding RyR2 luminal regulation. We believe these changes have strengthened the manuscript and enhanced its accessibility to the broader cardiac physiology community. Please find below a point-to-point response to each reviewer comment, indicating how and where the manuscript has been revised accordingly.
Reviewer 1
The present review by Marabelli et al., entitled “The Yin and Yang of Heartbeats: Magnesium-Calcium Antagonism is Essential for Cardiac Excitation–Contraction Coupling”, focuses on the roles played by magnesium ions (Mg2+) in cardiac excitation-contraction coupling process. The review nicely summarizes different steps of EC-coupling magnesium competes with calcium in fine tune regulation of action potential generation, calcium-induced calcium release, sarcoplasmic reticulum regulation, modulation of contractile myofilaments, and calcium extrusion during relaxation. The manuscript provides a detailed and comprehensive description of structural, biophysical, molecular, and electrophysiological mechanisms involved in magnesium-calcium antagonism. This review is timely; the manuscript is well-written and provides an objective overview of current literature on this topic. I just have a few minor suggestions and concerns listed below.
- It would be beneficial for readers to add one sentence at the end of the abstract concluding the main purpose/scope of the present review as it will appear on PubMed.
We thank the reviewer for pointing this out. We have adapted the abstract to include the following sentence: “This review aims to provide a comprehensive framework to link the structural and molecular mechanisms of Mg²⁺/Ca²⁺ antagonistic binding across key proteins of the cardiac ECC machinery to their physiopathological relevance.” (lines 15-18 in the rewritten manuscript).
- The review would benefit from including a summary schematic combining all the key effects of magnesium on EC-coupling steps. This can be shown as a schematic or using a classic cartoon by Donald Bers representing spatial arrangement of EC-coupling and contraction steps in a micro-domain specific manner.
We thank the reviewer for the useful suggestion. Because the effects of Mg2+ are quite complex and the interactions among ECC components are difficult to summarize, we have focused on drawing a simplified scheme summarizing the effects of hypomagnesemia on the ECC process, which has now been included in the rewritten manuscript as Figure 3, and is referred to at lines 724 and 738 (within the newly added section 6, see below).
- It might be reasonable to add a separate section at the end of the manuscript that will summarize the role of magnesium in the regulation of EC-coupling and, potentially, cardiac arrhythmias in the pathological settings, such as heart failure.
We recognize the value of addressing the clinical relevance of Mg2+ homeostasis and addressed this point by adding a new, clinically oriented final paragraph (section number 6), entitled “Pathological Dysregulation of Mg²⁺ and Therapeutic Implications” (see lines 706-743). In this section, we specifically discuss the effects of Mg²⁺ in ischemia, heart failure, and arrhythmias, emphasizing the physiological consequences of dysregulated Mg²⁺ handling.
- Similarly, it will be beneficial to add a brief section highlighting clinical relevance and potential therapeutic potential of magnesium in treating abnormal cardiac contraction/altered EC-coupling and, potentially, cardiac arrhythmias, such as torsades de pointes where magnesium is considered as the first-line treatment option.
We recognize the relevance of Mg2+ in the prevention and treatment of cardiac arrhythmias. Following the reviewer’s suggestion, we now explicitly highlight the therapeutic role of Mg2+ in the newly added section 6, lines 709-712. We write: “Mg²⁺ is one of the few ions used as a first-line treatment in emergency cardiovascular care: intravenous magnesium sulfate is administered for Torsades de pointes and other ventricular arrhythmias, and shows potential in long-term heart failure and hypertension management, and as a metabolic stabilizer in diabetes [10].”. However, in keeping with the structural and mechanistic scope of the review, we refrain from extensive discussion of clinical guidelines and refer the reader to organism-level reviews of Mg²⁺ homeostasis (see line 55).
- The authors should consistently use “AP” as an abbreviation for “action potential” once it is introduced at the beginning of the manuscript. The same is true for all other abbreviations. For example: Line 240 – There is no need to re-introduce ECC again.
We thank the reviewer for his attention to detail. We have carefully revised and harmonized the use of abbreviations throughout the manuscript to ensure consistency.
- Lines 284-287: In Ref. 60, the authors showed that “decreases in the intracellular Mg(2+) concentration, to levels outside their normal ranges prolonged action potential duration by decreasing the I(K1) transient.” Indeed, clinically, hypomagnesemia is associated with torsade de pointes which relies on APD prolongation and development of EADs. Hin contrast, as currently stated in line 287, APD shortening would rather suppress EAD formation. Please, check this.
We are sincerely thankful to the reviewer for pointing out this important inaccuracy. The revised sentence (at lines 286-289) now reads:
“The study concluded that lack of Mg2+ in these simulated conditions would unexpectedly decrease outward IK1 currents, leading to prolonged AP duration (see figure 5 therein), which in turn could result in increased risk of arrhythmias (early after-depolarizations, heterogeneity of AP duration) [60]”.
- Line 385: “RyR21” – should be “RyR1”.
We thank the reviewer for bringing this typographical error to our attention. The incorrect reference to “RyR21” has been corrected to “RyR1” in the revised manuscript (see line 387).
- Line 487: What is about another CaMKII-dependent phosphorylation side 2808?
We thank the reviewer for highlighting this critical detail capturing the complexity of RyR2 regulation. During β-adrenergic stimulation, both Ser-2808 and Ser-2814 on RyR2 are phosphorylated, and their close spatial proximity suggests a synergistic impact on channel activity. However, data on the specificity of CaMKII versus PKA at Ser2808 remain conflicting, as slight variations in experimental conditions led to contrasting results. Early studies indicated that canine RyR2-Ser2809 is phosphorylated by CaMKIIδ and by PKA to a lesser extent (Witcher et al., 1991, PMID: 1645727). Later studies showed CaMKIIδ uniquely phosphorylated Ser2815 (rodent 2814) of a recombinantly purified RyR2 (Wehrens et al., 2004, PMID: 15016728). To maintain clarity and accuracy, we took a conservative approach in the main text, describing only well-validated sites with clearly assigned kinases. Following the reviewer’s suggestion, we now include Ser-2808 in the discussion at lines 488-492: “However, when pathologically hyperactive, CaMKIIδ is involved in the pro-arrhythmogenic effects of catecholaminergic stimulation in the heart, in part mediated by phosphorylation of the target RyR2 residues Ser2814 (Ser2815 in large mammals) and Ser2808 (Ser2809 in the tested canine model), which enhances RyR2-mediated diastolic Ca2+ release [100, 101].”
The complete point to point responses to both reviewers can be found in the attached file.

Reviewer 2 Report
Comments and Suggestions for Authors
The manuscript of Marabelli et al. aims to summarize the role of Mg2+/Ca2+ antagonism in the regulation of cardiac excitation–contraction coupling. Mg2+ ions, the most abundant divalent cations in the cytoplasm, are essential for numerous cellular processes. As highlighted by the authors, Mg2+ often acts in opposition to the activating effects of Ca2+. Although Mg2+ has a relatively low affinity for protein binding sites, its cytosolic concentration—several orders of magnitude higher than that of free Ca2+- enables it to effectively compete with Ca2+ at Ca2+-activation sites, thereby exerting a critical modulatory influence. The topic of this review is compelling and may attract a broad readership within the cardiac research community. However, upon reading the manuscript, I identified several issues that should be addressed before it can be considered for publication.
Specific comments:
- Abstract: To improve clarity, please introduce the abbreviation “RyR2” by writing “cardiac ryanodine receptor (RyR2)” the first time it appears in the abstract.
- Lines 346-348: The authors state that the Cav1.2 channel lies within tens of nanometers from the mouth of the RyR2 channel. However, the typical size of cardiac dyads is approximately 15 nm. How, then, could Cav1.2 and RyR2 channels be separated by tens of nanometers?
- Line 364: I recommend that the authors cite the original experimental study by Li et al., 2013 (doi:10.1371/journal.pone.0058334) instead of Iaparov et al., 2025 (doi:10.3389/fphys.2021.805956), which does not specifically address the effect of RyR2 channel phosphorylation.
- Line 522: I found some potentially misleading information. The luminal L site on the RyR2 channel is activatory for luminal Ca²⁺, while Mg²⁺ is thought to inhibit RyR2 by competing with Ca²⁺ at this site, as proposed by Walweel et al., 2014 (doi:10.1085/jgp.201311157). Furthermore, Magyar et al., 2023 (doi:10.1016/j.bpj.2023.07.029) identified, in addition to the L site, an inhibitory site for luminal Ca²⁺. The effect of Mg²⁺ at this inhibitory site has not been tested. These points should be clarified in the revised manuscript.
- Line 577: Regarding the EF-hand-like pattern identified specifically within the first intraluminal loop of the RyR2 channel, the citation provided is incorrect. This information was reported in Gaburjakova and Gaburjakova, 2016 (doi:10.1016/j.bioelechem.2016.01.002), not in Gaburjakova and Gaburjakova, 2010 (doi:10.1007/s00232-010-9243-8). The citation should be corrected accordingly.
- Section 5.3.2. a. CASQ2: The work of Handhle et al. (doi: 10.1242/jcs.191643) should also be mentioned, as it highlights a potential direct interaction site between the RyR2 channel and CSQ2.
- Table 1, Protein location column: Appropriate citations should be provided. It would also be helpful if the authors indicate whether the reported location is supported by structural studies or is merely a predicted or suggested location. Importantly, the information regarding luminal sites on the RyR2 channel should be revised in accordance with the comments provided in points 4 and 5.
- Table 1, Divalent binding properties column: I do not understand why the authors included the binding affinity for Eu3+ under a column titled "Divalent Binding Properties." Europium (Eu3+) is a trivalent ion, not divalent, and therefore its inclusion appears inconsistent. Additionally, the citation of Nayak et al., 2024 (doi:10.1038/s41467-024-48292-3) is not appropriate in the context of selectivity filter properties and Eu3+ binding, as I could not find any mention of Eu3+ in that paper. Furthermore, there are numerous publications that specifically address the divalent ion binding properties of the RyR2 channel's selectivity filter, which should be considered instead.
Author Response
We thank both reviewers for their thoughtful and constructive comments, which greatly contributed to improve the quality and clarity of the manuscript. We appreciate the positive evaluation of the relevance and timeliness of this review study on the moonlighting role of Mg2+ in cardiac function. Following the insightful suggestions for refinement, we included now a schematic summary (Figure 3) to visually integrate the effects of Mg2+ on ECC components, and a new clinically-focused section summarizing the key pathological implications and therapeutic potential of magnesium in cardiac diseases. We have made several revisions to key paragraphs and Table 1 to improve precision and acknowledge recent findings, particularly regarding RyR2 luminal regulation. We believe these changes have strengthened the manuscript and enhanced its accessibility to the broader cardiac physiology community. Please find below a point-to-point response to each reviewer comment, indicating how and where the manuscript has been revised accordingly.
Reviewer 2
The manuscript of Marabelli et al. aims to summarize the role of Mg2+/Ca2+ antagonism in the regulation of cardiac excitation–contraction coupling. Mg2+ ions, the most abundant divalent cations in the cytoplasm, are essential for numerous cellular processes. As highlighted by the authors, Mg2+ often acts in opposition to the activating effects of Ca2+. Although Mg2+ has a relatively low affinity for protein binding sites, its cytosolic concentration—several orders of magnitude higher than that of free Ca2+- enables it to effectively compete with Ca2+ at Ca2+-activation sites, thereby exerting a critical modulatory influence. The topic of this review is compelling and may attract a broad readership within the cardiac research community. However, upon reading the manuscript, I identified several issues that should be addressed before it can be considered for publication.
- Abstract: To improve clarity, please introduce the abbreviation “RyR2” by writing “cardiac ryanodine receptor (RyR2)” the first time it appears in the abstract.
We thank the reviewer for this helpful observation. We have revised the abstract to introduce the abbreviation appropriately (see line 21). The term now appears as “cardiac ryanodine receptor (RyR2)” upon first mention, in accordance with standard scientific conventions.
2. Lines 346-348: The authors state that the Cav1.2 channel lies within tens of nanometers from the mouth of the RyR2 channel. However, the typical size of cardiac dyads is approximately 15 nm. How, then, could Cav1.2 and RyR2 channels be separated by tens of nanometers?
We have corrected the statement to more accurately reflect the known spatial organization of the cardiac dyad. The revised sentence, now appearing at line 380, reads: “within a dozen of nanometers”, which better aligns with the typical dimensions (8 to 20 nm) reported in the literature. In support of this correction, we have also added the recent study by Bäuerlein et al. (2023) as a new reference (reference number 78).
3. Line 364: I recommend that the authors cite the original experimental study by Li et al., 2013 (doi:10.1371/journal.pone.0058334) instead of Iaparov et al., 2025 (doi:10.3389/fphys.2021.805956), which does not specifically address the effect of RyR2 channel phosphorylation.
We appreciate the reviewer’s valuable recommendation. As suggested, we have replaced the previous citation with the original experimental work by Li et al., 2013, which more directly supports the point regarding RyR2 phosphorylation. The new reference is numbered 81, and in this revision it appears at line 366. The numbering of references has been updated accordingly throughout the manuscript
4. Line 522: I found some potentially misleading information. The luminal L site on the RyR2 channel is activatory for luminal Ca²⁺, while Mg²⁺ is thought to inhibit RyR2 by competing with Ca²⁺ at this site, as proposed by Walweel et al., 2014 (doi:10.1085/jgp.201311157). Furthermore, Magyar et al., 2023 (doi:10.1016/j.bpj.2023.07.029) identified, in addition to the L site, an inhibitory site for luminal Ca²⁺. The effect of Mg²⁺ at this inhibitory site has not been tested. These points should be clarified in the revised manuscript.
We sincerely thank the reviewer for this constructive comment, which has prompted a revision of the relevant section of the manuscript (3.2.2.2) to better clarify the distinct regulatory effects of luminal Mg²⁺ on RyR2, accounting for: i) the confounding role of cytosolic Ca2+ in masking luminal-specific effects and ii) the presence of accessory proteins. Furthermore, we now explicitly refer to the study by Magyar et al., 2023, which identified an additional inhibitory site for luminal Ca²⁺ distinct from the L-site. As correctly noted by the reviewer, the potential effect of Mg²⁺ at this second site has not yet been explored.
More specifically, we now anticipate to the readers that the Mg2+ -dependent, competitive inhibition of the Ca2+-dependent activation of RyR2 (described by Walweel at al., 2014) relies on the presence of protein partners, in the first introductory paragraph on the effect of lumimnal Mg2+, at lines 524-528:
“Experimental evidence, primarily obtained using RyR2 samples preserving associated luminal proteins such as calsequestrin (CASQ2), triadin (TRDN), and junctin (JNT), unveiled the role of the so-called L-site, an activation site for luminal Ca²⁺, at which luminal Mg²⁺ competes as a non-activating antagonist [12,115].”
and at lines 537-541:
“In the following paragraphs we will delve into how luminal Mg2+ regulates the open probability of RyR2 either in presence or absence of its soluble accessory protein CASQ2. Intriguingly, these two mechanisms, featuring distinct selectivities for Ca2+ Vs. Mg2+ ions, and different magnitudes of their physiological impacts, also feature opposite functional outcomes on RyR2.”.
Walweel et al. (2014), indeed used “native RyR2” from heart microsomes reconstituted into bilayers. Their experimental methods, as acknowledged by Laver and Honen (2008), “appear to retain their association with luminal proteins such as CASQ2, triadin, and junctin… The Ca²⁺/Mg²⁺ mechanisms explored should encompass the regulatory actions of these co-proteins.”
Conversely, paragraph 5.3.1 (lines 574-602), addressing the effect of Mg2+ on intrinsic RyR2 sites has been rephrased to better clarify the absence of CASQ2 in the experimental setups studied by Qin et al, 2008; Diaz-Sylvester et al, 2011; Gaburjakova et al., 2016; and Magyar et al, 2023.
Regarding the second part of the reviewer’s comment (Magyar et al. 2023), we now explicitly distinguish here between the effect of Mg2+ at:
- The intrinsic activatory RyR2 L-site (non-selective for Ca²⁺ and Mg²⁺, with an EC50 below the physiological concentration of lumninal Mg2+, and only revealed when CSQ2 is removed) (paragraph 5.3.1.). See lines 575-580:
“ When RyR2 is stripped of CASQ2, maximally activating concentrations of cytosolic Ca2+ (100 mM), unmask an intrinsic luminal Mg2+-activation mechanism, where luminal Mg2+ is equally efficient as luminal Ca2+ in sustaining RyR2 channel opening, with an EC50 for Ca2+/Mg2+ activation of 379 ± 247 µM [27,118]. More recently, Magyar et al., by probing at diastolic cytosolic Ca2+ (100 nM) levels with a non-permeating ion (Eu3+), corroborated the existence of an intrinsic luminal Ca2+-activating site in canine RyR2 [115]”.
- The intrinsic inhibitory L-site proposed by Magyar et al., 2023 (paragraph 5.3.1.). See lines 595-602:
“Recent studies also unveiled an additional, intrinsic, RyR2 luminal Ca2+-binding site or Ca2+-dependent, allosteric mechanism, which becomes apparent only under particular cytosolic Ca2+ concentrations, and is otherwise of questionable functional importance [115]. The location of this site has been proposed to be conserved between isoforms: close to the channel gate and the selectivity filter. However, there are currently neither functional nor structural data on the effect here played by Mg2+: while cryo-EM analysis of RyR1 shows an axial, non-protein density in the conserved selectivity filter region, its density did not increase with Mg2+ concentration [85]”.
The intrinsic luminal inhibtory site has also been added to Table 1.
To improve clarity, the effect of Mg2+ on the physiological RyR2 complex with CASQ2, TRDN, and JNT, which sees the competition between Ca²⁺-dependent activation and Mg²⁺ -dependent inhibition (paragraph 5.3.2.) has also been rephrased at lines 604-616:
“Since the effects of luminal Mg2+ on RyR2 in the absence of CASQ2 are not specific between Ca2+ and Mg2+, and their physiological meaning is only evident in special circumstances, the presence of luminal protein partners is the missing link to mecha-nistically explain the well-known, physiologically relevant, RyR2 inhibition by luminal Mg2+ (Figure 2). Partners ensure that the RyR2 is capable of discriminating between luminal Ca2+ (which activates RyR2 when interacting with partners) and luminal Mg2+ (crucial to prevent excessive Ca²⁺ release particularly during diastole, when cytoplasmic Ca2+ concentrations are low and the SR is well loaded with Ca2+) [12,87,96]. Furthermore, in a context where the physiological concentration of luminal Mg2+ is considered to stably remain at approximately 1mM (due to the absence of well-established active Mg²⁺ transport across the SR membrane), the discrimination between divalent cations is also necessary to ensure that the RyR2 activity is coupled with physiological fluctuations in the luminal free Ca2+ content between 0.3 and 1 mM [3,87]”.
We thank the reviewer again for his/her helpful observation, which improved the precision of our discussion.
5. Line 577: Regarding the EF-hand-like pattern identified specifically within the first intraluminal loop of the RyR2 channel, the citation provided is incorrect. This information was reported in Gaburjakova and Gaburjakova, 2016 (doi:10.1016/j.bioelechem.2016.01.002), not in Gaburjakova and Gaburjakova, 2010 (doi:10.1007/s00232-010-9243-8). The citation should be corrected accordingly.
We thank the reviewer for his/her correction. We have now updated the citation as suggested: the reference to Gaburjakova and Gaburjakova, 2010 has been replaced with the correct one, Gaburjakova and Gaburjakova, 2016 which more accurately supports the statement regarding the EF-hand-like pattern in the first intraluminal loop of the RyR2 channel. This correction is reflected in citation number 118 of the revised manuscript (see line 578).
6. Section 5.3.2. a. CASQ2: The work of Handhle et al. (doi: 10.1242/jcs.191643) should also be mentioned, as it highlights a potential direct interaction site between the RyR2 channel and CSQ2.
We thank the reviewer for this valuable suggestion. We fully agree that the study by Handhle et al. represents an important contribution, as it provides evidence for a potential direct interaction site between the RyR2 channel and calsequestrin (CSQ2). We have now incorporated this reference into the revised manuscript at line 652, and it is cited as reference number 132.
7. Table 1, Protein location column: Appropriate citations should be provided. It would also be helpful if the authors indicate whether the reported location is supported by structural studies or is merely a predicted or suggested location. Importantly, the information regarding luminal sites on the RyR2 channel should be revised in accordance with the comments provided in points 4 and 5.
We sincerely thank the reviewer for this important and constructive suggestion. We have now revised the “Protein location” column in Table 1 to clarify whether the proposed site is experimentally demonstrated (e.g., via mutagenesis or trypsinization), structurally resolved, or rationally/computationally predicted. Corresponding references have been included, and a standardized notation has been introduced to indicate the level of validation.
As detailed in the revised table legend (lines 794–798), the references supporting the localization of functional sites have been categorized as follows:
a – based on homology with RyR1, for which structural evidence is reported;
b – based on mutagenesis or trypsinization studies;
c – based on rational or computational hypotheses.
In addition, we have expanded the table to include the “Intrinsic Luminal Inhibition Site”, corresponding to the Ca²⁺-dependent inhibitory site described by Magyar et al., 2023. This site is also discussed in the revised main text, in line with the reviewer's point 5.
8. Table 1, Divalent binding properties column: I do not understand why the authors included the binding affinity for Eu3+ under a column titled "Divalent Binding Properties." Europium (Eu3+) is a trivalent ion, not divalent, and therefore its inclusion appears inconsistent. Additionally, the citation of Nayak et al., 2024 (doi:10.1038/s41467-024-48292-3) is not appropriate in the context of selectivity filter properties and Eu3+ binding, as I could not find any mention of Eu3+ in that paper. Furthermore, there are numerous publications that specifically address the divalent ion binding properties of the RyR2 channel's selectivity filter, which should be considered instead.
We thank the reviewer for his/her careful evaluation and fully agree that the original wording and citation were misleading. Specifically, the inclusion of Eu³⁺ under a column titled “Divalent Binding Properties” was inappropriate, as Europium is a trivalent ion. We have changed the column title to “Ion Binding Properties” to more accurately reflect the broader scope of the data presented. We have replaced the incorrect citation of Nayak et al., 2024, which does not mention Eu³⁺, with that of Magyar et al., 2023, who employed this ion as a specific luminal agonist to probe Ca²⁺-dependent regulatory sites on RyR2, particularly in the context of avoiding feed-through artifacts during functional assays.
The complete point-to-point responses to both reviewers can be found in the attached file.
Round 2
Reviewer 2 Report
Comments and Suggestions for Authors
I appreciate the thorough revision made in response to my comments. The changes have improved the clarity and quality of the work. I have no further comments, and I recommend the manuscript for publication.